# Investigating the Determinants of Mortality before CD4 Count Recovery in a Cohort of Patients Initiated on Antiretroviral Therapy in South Africa Using a Fine and Gray Competing Risks Model

**DOI:** 10.3390/tropicalmed9070154

**Published:** 2024-07-10

**Authors:** Chiedza Elvina Mashiri, Jesca Mercy Batidzirai, Retius Chifurira, Knowledge Chinhamu

**Affiliations:** 1Department of Applied Mathematics and Statistics, Midlands State University, Gweru 9055, Zimbabwe; 2School of Mathematics, Statistics and Computer Science, University of KwaZulu-Natal, Howard College Campus, Durban 4041, South Africa; chifurira@ukzn.ac.za (R.C.); chinhamu@ukzn.ac.za (K.C.); 3School of Mathematics, Statistics and Computer Science, University of KwaZulu-Natal, Pietermaritzburg Campus, Pietermaritzburg 3209, South Africa; batidzirai@ukzn.ac.za

**Keywords:** Fine–Gray model, competing risk, mortality, CD4 count recovery, cumulative incidence

## Abstract

CD4 count recovery is the main goal for an HIV patient who initiated ART. Early ART initiation in HIV patients can help restore immune function more effectively, even when they have reached an advanced stage. Some patients may respond positively to ART and attain CD4 count recovery. Meanwhile, other patients failing to recover their CD4 count due to non-adherence, treatment resistance and virological failure might lead to HIV-related complications and death. The purpose of this study was to find the determinants of death in patients who failed to recover their CD4 count after initiating antiretroviral therapy. The data used in this study was obtained from KwaZulu-Natal, South Africa, where 2528 HIV-infected patients with a baseline CD4 count of <200 cells/mm^3^ were initiated on ART. We used a Fine–Gray sub-distribution hazard and cumulative incidence function to estimate potential confounding factors of death, where CD4 count recovery was a competing event for failure due to death. Patients who had no tuberculosis were 1.33 times at risk of dying before attaining CD4 count recovery [aSHR 1.33; 95% CI (0.96–1.85)] compared to those who had tuberculosis. Rural patients had a higher risk of not recovering and leading to death [aSHR 1.97; 95% CI (1.57–2.47)] than those from urban areas. The patient’s tuberculosis status, viral load, regimen, baseline CD4 count, and location were significant contributors to death before CD4 count recovery. Intervention programs targeting HIV testing in rural areas for early ART initiation and promoting treatment adherence are recommended.

## 1. Introduction

The provision of antiretroviral therapy (ART) to HIV-infected individuals has saved millions of lives in resource-limited settings since HIV has remained a global health concern [1,2]. Globally in 2022, 39.0 million were living with HIV, 1.3 million people became newly infected with HIV, and 630,000 people died from AIDS-related illnesses [3]. ART has led to low levels of morbidity and mortality, immunological recovery, and viral load reduction [4,5,6,7]. Immunological recovery or CD4 count recovery is when the patient’s CD4 count has reached more than 500 cells/mm^3^. About 15–20% of patients who initiate ART with very low CD4 counts (200 cells/mm^3^) may continue to have abnormally low CD4 cell counts and have the highest risk of failure to achieve optimal immunological recovery [8,9]. Patients on ART with poor CD4 recovery early in treatment are at greater risk of progression to new AIDS diagnosis or death despite viral suppression.

CD4 cell counts remain low among individuals presenting for care in many areas [10] and up to 17% of adults die in sub-Saharan Africa in the first year after ART initiation [11]. Consequently, in developing countries, high mortality rates persist despite the availability of antiretroviral and AIDS-defining events that continue to be reported as the main cause of hospitalization and death [12,13,14,15,16]. Risk factors for death in the initial year after starting ART have been linked to a low CD4 count < 50 cells/mm^3^ conferring particularly high risk [17]. In some cases, drug resistance may also cause an abnormally low CD4 count. Individuals with drug-resistant strains of HIV may experience treatment failure, leading to disease progression and an increased risk of mortality.

Patients with advanced-stage disease, for example, WHO clinical stage 3 or 4, may have a higher risk of mortality before CD4 count recovery due to more severe illness and opportunistic infections [18]. Rapid disease progression or delayed diagnosis in patients may have more extensive immune damage, making them more susceptible to complications and death before CD4 count recovery [19]. Studies on competing events associated with patients initiated on ART were done by different authors. Yang et al. (2021) researched on competing risks of CD4 count and AIDS-related mortality. The results showed that low CD4 was associated with higher AIDS-related mortality. Patients with CD4 count ≥ 500 cells/mm^3^ at baseline had lower AIDS-related mortality [20]. Progression rates for AIDS and non-AIDS are significantly the same. However, patients who initiate ART with a CD4 count < 200 cells/mm^3^ are at risk even after CD4 count recovery [21]. There is also a high risk of loss to follow-up and mortality in patients with low baseline CD4 count [22,23]. Patients with high CD4 counts who initiate ART late had a higher risk of AIDS-related death than those with low CD4 counts who start ART early [20]. Our research objective was to find the determinants of death before CD4 count recovery in the presence of competing events. Out of the patients who had not reached a CD4 count of at least 500 cells/mm^3^, death precluded the occurrence of CD4 count recovery, hence competing risk models were implemented. This study implores competing risk analysis techniques to determine the time taken to CD4 count recovery as well as to identify the determinants of death before CD4 count recovery for people living with HIV/AIDS infection under ART in KwaZulu-Natal, South Africa.

## 2. Materials and Methods

### 2.1. Study Setting and Population

The study’s two locations, eThekwini and Vulindlela, are urban and rural, respectively. Patients who missed three or more visits in a row and failed to be physically tracked were recorded as lost to follow-up and excluded in the study.

### 2.2. Study Design and Data

A retrospective cohort study was conducted among HIV-infected patients initiating ART at the South African Centre for the AIDS Program of Research (CAPRISA). The study used a non-probability sampling technique called convenience sampling. This study used data from the South African Centre for the AIDS Program of Research (CAPRISA). The dataset was given to the author on 17 June 2022. No authors could identify the patients because the data set used unique patient identification. The program enrolled 4013 patients for antiretroviral therapy initiation between June 2004 and August 2013. In our study, 2528 patients met the 2004 eligibility criteria of CD4 count below 200 cells/mm^3^ or WHO stage 4 AIDS-defining illness [24]. Between 2006 and 2009, this criterion was raised to 350 cells/mm^3^ [25].

### 2.3. Study Variables

CD4 counts, viral load, demographic characteristics, and medication regimens were recorded at baseline. The standard first-line ART treatment was two nucleoside reverse transcriptase inhibitors and a non-nucleoside reverse transcriptase inhibitor. Second-line treatment was comprised of a protease inhibitor (PI) plus two nucleoside analogs (NRTIs) and is suitable for unsuccessful first-line treatment. CD4 count and viral load were taken at the beginning and every six months or as needed for clinical reasons.

### 2.4. Statistical Analysis

Patients’ characteristics were summarized using descriptive statistics including median and interquartile range for continuous covariates and frequencies for categorical measures. The Fisher’s Exact test was used to find the relationship among categorical variables. The *t*-test and Wilcoxon Signed rank test were used to explore the association between continuous variables, with *p*-values less than 0.2 implying the significance of the variables. All the association tests used a two-sided test. Fine and Gray competing risk regression [26] was used to calculate sub-distribution hazard ratios (sHR) with 95% confidence intervals for the different factors. Competing events from ART initiation were CD4 count recovery and mortality before CD4 count recovery. The cumulative incidence function was used to assess the effects of the covariates. We performed bivariate and multivariable competing regression analyses on the baseline characteristics. Variables from the bivariable model with a *p*-value less than 0.2 were included in the multivariable model. R version 4.1.2 comprisk package was used to perform the statistical analysis.

### 2.5. Statistical Model Formulation

#### 2.5.1. Competing Risk Model

A competing risk model is a statistical framework used in survival analysis to account for the presence of multiple types of events or outcomes that compete with each other. In many real-life situations, individuals may experience multiple events, and the occurrence of one event may preclude the occurrence of others. Competing risks arise when the event of interest for example death from a specific cause is influenced by the presence of other events like CD4 count recovery or loss of follow-up. The competing risk model allows for the estimation and analysis of the cumulative incidence function (CIF) for each event type, considering the presence of competing events. It provides a more accurate understanding of the probabilities and risks associated with each event type, accounting for the fact that not all individuals will experience the event of interest due to the occurrence of competing events. Statistical methods used in competing risk analysis may include the Fine–Gray model. This method allows for the estimation and comparison of cumulative incidence function as well as the assessment of covariates effects on the occurrence of the event of interest in the presence of competing risks. The provision of a more comprehensive understanding of the risk profile and event probabilities when accounting for the complexities introduced by competing events is well defined.

#### 2.5.2. The Cumulative Incidence Function (CIF)

The cumulative function is a commonly used statistical model in epidemiology and survival analysis to estimate the probability of a specific event occurring over time, given a specific risk factor. The CIF model is used to analyze the occurrence of a specific event, such as a disease or death, in a population over a specific period. The CIF can be defined as the probability of a specific event occurring within a specific time interval, given that the subject has not experienced the event before the start of the interval. The CIF is calculated using the following formula:(1)$$CIF\;(t)=\frac{P\;(T\leq t,E=1)}{P\;(E=0)}$$
where t is the time interval, T is the time to the occurrence of the event, and E is the event of interest. P(T≤t,E=1) is the probability that the event of interest occurs within the time interval, and P(E=0) is the probability that the event of interest does not occur before the start of the interval.

#### 2.5.3. Fine–Gray Model

The Fine–Gray [26] competing risk model was developed to consider competing risks and provide an estimation for the event of interest involving one or more competing risks. It is an extension of the Cox proportional hazards model and considers the cumulative incidence of different types of events while accounting for the competing nature of the risks. It allows for the estimation of the sub-distribution hazard ratio, which represents the effect of a covariate on the cumulative incidence function of a particular event, in the presence of competing risks. Fine and Gray formulated the sub-distribution hazard rate for event type k at time t individuals that failed from an event other than k before t remain in the risk set. The model is given by
(2)$$\gamma_{k}\left(t\right)=\underset{∆t\to 0}{\lim}\frac{P(t\leq T<t+∆t,D=k|T>t\cup \{T<t,D\neq k)}{∆t}$$

Equation (2) estimates the hazard rate for event k at time t on the risk set based on the risk set that remains at time t after accounting for all previously occurring event types, which includes competing events.

## 3. Results

Table 1 presents the baseline and follow-up characteristics of the 2528 HIV patients who initiated ART with CD4 < 200 cells/mm^3^ according to the 2004 eligibility criteria [24]. Of these, 727 (29%) achieved a CD4 count of at least 500 cells/mm^3^, 1498 (59%) had a CD4 count of less than 500 cells/mm^3^, and 303 (12%) died before recovery. Females who did not recover were 869 (35%), 178 (7%) died before CD4 count recovery, and 542 (21%) recovered their CD4 count. Age groups for men and women ranged from 14 to 76 years with *p* < 0.001 and a median of 32 and inter-quartile range of 28–39 from those who died before CD4 count recovery. The table also shows that a maximum number of deaths were recorded from patients with WHO Stage 3, 175 (7%) followed by WHO Stage 4, 62 (2%), WHO Stage 2, 42 (2%), and WHO Stage 1, 22 (1%). Patients without tuberculosis and had not recovered their CD4 count were 1153 (46%), 594 (24%) recovered their CD4 count and 257 (10%) died before reaching a CD4 count of more than 500 cells/mm^3^. Rural patients who died before recovering their CD4 were 209 (8%), 557 (22%) had a CD4 count of more than 500 cells/mm^3^, and 699 (28%) did not recover their CD4 count. Urban patients with less than 40 years had a 71 (3%) risk of dying before CD4 count recovery compared to 23 (1%) of patients aged between 40 and 59. Patients with less than 40 years had 593 (23%) failing to recover their CD4 count. Patients from rural areas with an age of less than 40 had 158 (6%) of dying before CD4 count recovery and 544 (22%) not recovering their CD4 count. The median and interquartile range baseline CD4 count was 109 (52–154) for patients who recovered their CD4 count, 95 (48–145) for those who did not recover their CD4 count, and 62 (25–102) who died before recovery. Patients who died on first-line treatment were 223 (9%), 586 (23%) recovered, and 1225 (48%) did not recover their CD4 count. Patients who died with a viral load of more than 400 copies/mL were 194 (8%), 113 (5%) had a CD4 ≥ 500 cells/mm^3^, and 418 (17%) did not recover their CD4 count.

### 3.1. Cumulative Incidence Functions for Selected Factors

Figure 1 shows the Cumulative Incidence function for gender, tuberculosis prevalence, and location. Patients from the urban area had a lower cumulative incidence risk of dying before CD4 count recovery compared to rural patients. The HIV-infected patients from the rural areas who had not recovered their CD4 count had a lower risk in comparison to urban patients. Patients without tuberculosis showed a potentially high risk of dying compared to those with tuberculosis, whilst those with tuberculosis had a lower risk in the CD4 count recovered group. Females had a lower cumulative incidence risk of dying before CD4 count recovery than males. Males who had not recovered their CD4 count had a greater risk than females.

### 3.2. Factors Affecting Competing Events

Table 2 shows the Fine–Gray sub-distribution hazard model, which estimates the effect of covariates on the sub-distribution hazard function on the competing events. Significant variables with *p*-value < 0.2 from the bivariate analysis were site, regimen, viral load, baseline CD4 count, gender, WHO stage, and tuberculosis, which were used to perform the multivariate analysis. Patients who were not suffering from tuberculosis were 1.33 times at risk of dying before reaching a CD4 count of more than 500 cells/mm^3^ [aSHR 1.33; 95% CI (0.96–1.85)] compared to the prevalent tuberculosis patients. The analysis showed that rural patients had a significantly higher risk of not recovering CD4 count and leading to HIV-related death [aSHR 1.97; 95% CI (1.57–2.47)] compared to urban patients. The baseline CD4 count had an increased association with death before recovery of patients [aSHR 0.99; 95% CI (0.99–1)]. Patients with a viral load of more than 400 copies/mL had a 72% risk of dying before CD4 count recovery than those with at most 400 copies/mL [aSHR 0.18; 95% CI (0.14–0.23)]. Second-line treatment patients had a significantly higher risk of dying before CD4 count recovery compared to first-line treatment patients [aSHR 1.84; 95% CI (1.45–2.34)]. Females had a positive association with mortality before CD4 count recovery [aSHR 1.18; 95% CI (0.98–1.49)] than males.

## 4. Discussion

This study used competing risk models to determine factors that contribute to the death of patients who were initiated on antiretroviral before reaching a CD4 count of at least 500 cells/mm^3^, those who did not experience CD4 count recovery and those who recovered their CD4 count. Our results showed a low rate of 29% CD4 count recovery for patients who initiated ART with less than 200 cells/mm^3^ and 59% failed to recover their CD4 count. The statistics found showed that it is recommendable to initiate ART early or with a CD4 count of more than 200 cells/mm^3^. Patients with no tuberculosis had a higher risk of mortality compared to the prevalent tuberculosis. The results reaffirm the results found by other authors [27,28]. Prevalent tuberculosis showed a low risk of mortality because they are initiated early in ART, and co-infections like tuberculosis are identified at an early stage. Immunological restoration through antiretroviral therapy reduces morbidity and mortality in prevalent tuberculosis patients [29]. HIV patients with low CD4 counts are at high risk of developing tuberculosis compared to individuals without HIV infection. If the level CD4 count of a patient is low, then the ability of the immune system to fight against infections is compromised, leading to individuals being susceptible and transitioning from HIV to AIDS [30,31]. The cumulative incidence of patients with prevalent tuberculosis was lower than those with tuberculosis in patients with recovered CD4 count. Effective tuberculosis treatment helps cure the infection, leading to the reduction of tuberculosis reactivation or acquiring new tuberculosis infections. Prevalent tuberculosis patients had a low risk of developing active tuberculosis compared to those without tuberculosis, since the immune system regained strength and can effectively control tuberculosis infections [32].

Rural patients showed an elevated risk of mortality before recovery of CD4 count than their urban counterparts. Rural areas have inferior health facilities and limited resources and medication [33,34]. Patients having to walk long distances to get medical assistance, poor background presentation, and low levels of literacy lead to high rural mortality [35]. Patients in urban areas access their HIV medication conveniently, since there are many health facilities at their disposal. Rural patients had an increased cumulative incidence than urban patients who had recovered their CD4 count [36]. The contributing factors are socioeconomic factors, environmental factors, and social support networks. Rural patients with HIV may face challenges in terms of social support networks caused by isolation, stigma, and limited access to support groups or networks that impact mental health, treatment adherence, and overall well-being, potentially influencing cumulative incidence rates [36]. Rural areas may have higher levels of poverty, limited education, and fewer economic opportunities compared to urban areas [35]. These socioeconomic factors can influence health-seeking behavior, adherence to treatment, and overall health outcomes, which may contribute to variations in cumulative incidence. Rural areas may have different environmental conditions that can impact health outcomes. For example, exposure to agricultural chemicals, limited access to clean water and sanitation, or increased prevalence of certain co-infections can contribute to the increased risk of specific outcomes in rural patients with HIV. The availability of comprehensive HIV care, early diagnosis, consistent access to ART, and ongoing monitoring are crucial for reducing the burden of HIV-related outcomes in both rural and urban settings [37]. Availability of comprehensive HIV care, early diagnosis, consistent access to ART, and ongoing monitoring are crucial for reducing the burden of HIV-related outcomes in both rural and urban settings [38,39].

Females experienced a lower cumulative incidence than males when they had not recovered their CD4 count. The causes of the disparities are behavior, access to health care and treatment, and social and cultural factors. Differences in sexual behavior and risk-taking practices can influence the cumulative incidence of HIV-related outcomes. Males may engage in higher-risk sexual behaviors, such as having multiple sexual partners or engaging in unprotected sex, which can increase their risk of HIV acquisition and disease progression [40]. Females may have better access to health care services, including HIV testing, prevention measures, and treatment, which can contribute to lower cumulative incidence rates. Factors such as antenatal care utilization, which provides opportunities for HIV testing and prevention interventions, can play a role in reducing HIV incidence among females [41]. Gender norms, power dynamics, and societal expectations can influence HIV risk and outcomes. Cultural practices or norms that promote male dominance or discourage female empowerment can contribute to increased HIV risk for males [42]. Comprehensive HIV prevention strategies, access to testing and treatment, and addressing gender-specific barriers are crucial for reducing the burden of HIV-related outcomes in both males and females. Co-infections and comorbidities may be experienced in females when the cumulative incidence increases in females than males after CD4 count recovery. Sexually transmitted infections or gynecological conditions may be more prevalent in females and can influence HIV progression or the development of specific complications [43]. The results from the study showed that males had an increased cumulative incidence of mortality before CD4 count recovery compared to females. The results are supported by other authors who stated that males had higher patient attrition and mortality compared to females, and this may be attributed in part to late presentation for HIV treatment and care [44,45].

Patients on first-line medication had higher chances of experiencing death compared to the ones on second-line medication. Reports from other authors showed that patients who took too long to switch to the second-line medication due to drug resistance, virological failure, and rapid CD4 count decline had a high risk of dying [46] and transmitting HIV to uninfected sex partners [47,48,49]. Our results showed that patients with low viral load significantly had lower chances of death compared to those with higher viral load. The HIV viral load was considered a standard marker for the evaluation of treatment success and to detect virological failure. WHO defined virological failure as two consecutive viral loads of more than 1000 copies/mL after 6 or more months [8,50]. At ART initiation, the median CD4 count was 109 with an interquartile range of 52–154 for patients who recovered their CD4 count. Patients with baseline CD4 cell counts of less than 50 cells/mm^3^ have a greater chance of immunological recovery compared to those with higher CD4 counts. Consequently, their CD4 counts remain below 200 cells/mm^3^ for a greater period, leading to an increment in morbidity and mortality [51]. Several studies showed that baseline CD4 count is associated with immune reconstitution [52,53]. Immune reconstitution is not entirely determined by the baseline CD4 count but is also influenced by factors such as treatment adherence, duration of HIV infection, presence of comorbidities, age, and overall health status [54]. Monitoring CD4 counts and assessing immune reconstitution is an essential part of HIV care. Regular monitoring allows healthcare providers to evaluate the effectiveness of ART, assess the individual’s immune status, and make appropriate adjustments or modifications to the treatment regimen if necessary [55].

## 5. Conclusions

A competing risk model was used to study time to death before CD4 count recovery after ART initiation, in the presence of possible recovery. Results showed that the patient’s tuberculosis status, viral load, regimen, baseline CD4 count, gender, WHO stage, and location were significant contributors to death before CD4 count recovery. A baseline CD4 count is important for assessing HIV progression, predicting clinical outcomes, and monitoring treatment responses. Individuals with lower baseline CD4 counts may experience a slower or less CD4 count recovery compared to those with higher baseline values.

Implementation of programs and campaigns to promote HIV testing is recommended so that patients may initiate ART immediately after diagnosis of the disease. Delaying initiation of ART puts a patient at risk of dying and immune deterioration. In addition, rural health facilities may be improved to monitor possible drug resistance to reduce mortality levels and address social determinants of health to improve overall HIV care and outcomes. HIV patients, particularly those with low CD4 counts, must receive appropriate and timely tuberculosis screening, preventive therapy, and access to ART.

## Figures and Tables

**Figure 1 tropicalmed-09-00154-f001:**
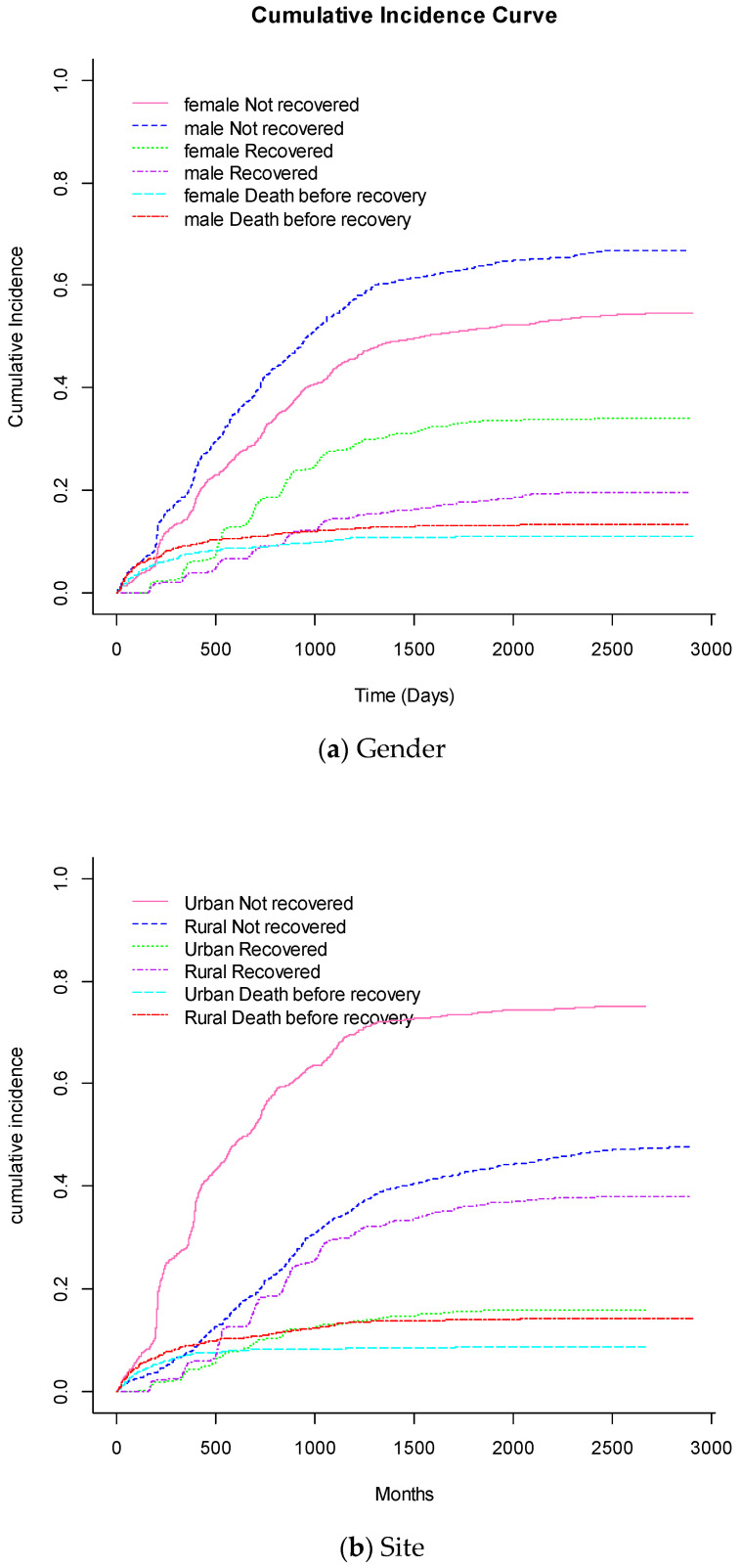
Non-parametric estimates of cumulative incidence curves with “Death before recovery”, “Recovered” and “Not recovered” as competing events for selected variable categories of (**a**) Gender, (**b**) Site, and (**c**) Tuberculosis.

**Table 1 tropicalmed-09-00154-t001:** Baseline Socio-demographic and clinical characteristics for ART-initiated patients between June 2004 and August 2013 in KwaZulu-Natal, South Africa.

Characteristics	Not Recovered CD4 Count (N = 1498) (59%)	Recovered CD4 Count (N = 727) (29%)	Death before CD4 Recovery (N = 303) (12%)	*p*-Value
Site, *n* (%)				
Urban (eThekwini)	799 (31%)	170 (7%)	94 (4%)	
Age < 40	599 (23%)	139 (5%)	71 (3%)	
40–59	199 (8%)	31 (1%)	23 (1%)	
>60	7 (0)	0	0	
Rural (Vulindlela)	699 (28%)	557 (22%)	209 (8%)	<0.001
Age < 40	544 (22%)	438 (17%)	158 (6%)	
40–59	145 (6%)	114 (5%)	47 (2%)	
>60	10 (0%)	5 (0%)	4 (0%)	
Age *, median (IQR)	34 (29–39)	32 (28–38)	32 (28–39)	<0.001
WHO Stage ** *n* (%)				
1	194 (8%)	105 (4%)	22 (1%)	
2	292 (12%)	136 (5%)	42 (2%)	<0.001
3	849 (34%)	419 (17%)	175 (7%)	
4	159 (6%)	65 (2%)	62 (2%)	
TB status, *n* (%)				
TB present	345 (14%)	133 (5%)	46 (1%)	0.001
No TB	1153 (46%)	594 (24%)	257 (10%)	
Viral load (copies/mL)				
>400	418 (17%)	113 (5%)	194 (8%)	
≤400	1080 (42%)	614 (24%)	109 (4%)	<0.001
Regimen				
Regimen 1 (First line)	1225 (48%)	586 (23%)	223 (9%)	0.006
Regimen 2 (Second line)	273 (11%)	141 (6%)	80 (3%)	
Gender, *n* (%)				
Female	869 (35%)	542 (21%)	178 (7%)	<0.001
Male	629 (25%)	185 (7%)	125 (5%)	
CD4 count median (IQR)	95 (48–145)	109 (52–154)	60 (25–102)	<0.001

* 3 missing age, ** 8 missing WHO stages.

**Table 2 tropicalmed-09-00154-t002:** Bivariate and multivariate competing risk regression analysis for predictors of mortality before CD4 count recovery among HIV patients on ART between June 2004 and August 2013 in KwaZulu-Natal, South Africa.

	Death before Recovery			
Characteristics	cSHR (95% CI)	*p*-Value	aSHR (95% CI)	*p*-Value
Gender (Ref: Male) Female	1.21 (0.96–1.52)	0.11 ***	1.18 (0.93–1.49)	<0.001 ***
Age	1 (0.99–1.01)	0.99		
TB (Ref: TB present) NO TB	1.47 (1.07–2.02)	0.02 ***	1.33 (0.96–1.85)	<0.001 ***
WHO stage 1	0.25 (0.06–1.04)	0.06 ***	0.24 (0.06–0.97)	<0.001 ***
WHO stage 2	0.33 (0.08–1.33)	0.12 ***	0.32 (0.08–1.25)	<0.001 ***
WHO stage 3	0.45 (0.11–1.81)	0.26 ***	0.44 (0.12–1.69)	<0.001 ***
WHO stage 4 (Ref)				
Site (Ref: Urban) Rural	1.65 (1.29–2.1)	<0.001 ***	1.97 (1.57–2.47)	0 ***
Regimen (Ref: First line treatment) Second line treatment	1.52 (1.18–1.97)	0.001 ***	1.84 (1.45–2.34)	0.01 ***
Viral load (Ref: ≤400 copies/mL) >400 copies/mL	0.17 (0.13–0.21)	<0.001 ***	0.18 (0.14–0.23)	<0.001 ***
Baseline CD4 count	0.99 (0.98–0.99)	0 ***	0.99 (0.99–1)	0.02 ***

*** means significant.

## Data Availability

CAPRISA has an established procedure to make its research data more broadly available. Information on the process for requesting and obtaining data is available on the CAPRISA website (www.caprisa.org) from 17 June 2022. Datasets used for analyses for the CAPRISA research article that has been published, can be requested by any investigator through an online request lodged on the CAPRISA website the CAPRISA Scientific Review Committee, and once approved the datasets, study protocol and statistical analysis plan will be made available to interested investigators making the request.

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
