# Peer review of "Investigating the Determinants of Mortality before CD4 Count Recovery in a Cohort of Patients Initiated on Antiretroviral Therapy in South Africa Using a Fine and Gray Competing Risks Model"

_tropicalmed, 2024, doi:10.3390/tropicalmed9070154_

Round 1
Reviewer 1 Report
Comments and Suggestions for Authors
In this study, Mashiri et al evaluated the determinants of death in a cohort of patients who failed to recover their CD4 counts after initiating antiretroviral therapy. Although the main results are just corroborating those from previous studies, this manuscript is interesting in the way it can be used to support the implementation of public health policies.
In this sense, a more straightforward discussion could be done:
Authors state that “CD4 count recovery is the main goal for an HIV patient who initiated ART.” Although this is true considering ART initiation only in an advanced stage of HIV infection, there are several protocols suggesting that it is quite important to initiate ART at the first diagnosis, independently of CD4 counts. I understand difficulties in follow such protocol in some situations, but I would like to see some discussion at this point.
In fact, a baseline CD4 count of <200 cells/𝑚𝑚3 (or even <350 cells/𝑚𝑚3) seems to be a quite low threshold to initiate ART, in my opinion.
Considering the authors conclusions “The patient’s tuberculosis status, viral load, regimen, baseline CD4 count, and location were significant contributors to death before CD4 count recovery.” (pay especial attention to baseline CD4 counts”), both above points deserve a better discussion.
Finally, authors conclude that “Implementation of programs and campaigns to promote HIV testing is recommended so that patients may initiate ART on time.”, though they did not discuss when this “on time” would be (a baseline CD4 count of <200 cells/𝑚𝑚3 as previously considered, <350 cells/𝑚𝑚3 as it seems to be the present indications, or any other moment?).
I believe that this is the main question of the manuscript, and deserves to be emphasized.
Comments on the Quality of English LanguageAttention should be given to some sentences, for example “Recent studies researched different…” and “Therefore, this study implores competing risk analysis techniques to determine the time taken to CD4 count recovery…”
Reviewer 2 Report
Comments and Suggestions for Authors
Overall this paper was well written and the analyses were done by an experienced statistician. The findings were not that impactful, but the study was still done well. Some minor suggested edits below:
· Abstract “Some patients respond positively to ART and attain CD4 count recovery, some patients might fail to recover their CD4 count due to non-adherence, treatment resistance, and virological failure, leading to HIV-related complications and death.” Please break this into two sentences
· Abstract “Patients who were not suffering from tuberculosis were 1.33 times at risk of dying before attaining CD4 count recovery [aSHR 1.33; 95% CI (0.96-1.85] compared to those who had no tuberculosis.” This sentence is confusing, please rewrite. Do you mean that patients with TB had a higher risk of dying compared to patients without TB?
· Line 43, indentation is too far to the right. Also with Lines 233 & 256. Check spacing in Line 279
· Table 1. It would be helpful to have the Urban and Rural Populations stratified by Age. If the age is significantly older in the Rural population, it would make sense that you find a higher risk of dying in this population.
· Line 180: Is there supposed to be text under the subheading “3.1. Baseline Patient Characteristics” or is this supposed to go before Table 1?
